# Lipid Profile Variations in Pregnancies with and without Cardiovascular Risk: Consequences for Both Mother and Newborn

**DOI:** 10.3390/children10091521

**Published:** 2023-09-07

**Authors:** Simona-Alina Abu-Awwad, Marius Craina, Lioara Boscu, Elena Bernad, Paula Diana Ciordas, Catalin Marian, Mircea Iurciuc, Ahmed Abu-Awwad, Stela Iurciuc, Brenda Bernad, Diana Maria Anastasiu Popov, Anca Laura Maghiari

**Affiliations:** 1Doctoral School, “Victor Babes” University of Medicine and Pharmacy, 300041 Timisoara, Romania; alina.abuawwad@umft.ro (S.-A.A.-A.); lioara.boscu@umft.ro (L.B.); 2Clinic of Obstetrics and Gynecology, “Pius Brinzeu” County Clinical Emergency Hospital, 300723 Timisoara, Romania; mariuscraina@hotmail.com (M.C.); bernad.elena@umft.ro (E.B.); 3Department of Obstetrics and Gynecology, Faculty of Medicine, “Victor Babes” University of Medicine and Pharmacy, 300041 Timisoara, Romania; 4Center for Laparoscopy, Laparoscopic Surgery and In Vitro Fertilization, “Victor Babes” University of Medicine and Pharmacy, 300041 Timisoara, Romania; 5Center for Neuropsychology and Behavioral Medicine, “Victor Babes” University of Medicine and Pharmacy, 300041 Timisoara, Romania; 6Departament IV—Discipline of Biochemistry, “Victor Babes” University of Medicine and Pharmacy, 300041 Timisoara, Romania; paulamuntean22@gmail.com (P.D.C.); cmarian@umft.ro (C.M.); 7Departament VI—Discipline of Outpatient Internal Medicine, Cardiovascular Prevention and Recovery, “Victor Babes” University of Medicine and Pharmacy, 300041 Timisoara, Romania; mirceaiurciuc@gmail.com (M.I.); iurciuc.stela@umft.ro (S.I.); 8Department XV—Discipline of Orthopedics—Traumatology, “Victor Babes” University of Medicine and Pharmacy, 300041 Timisoara, Romania; ahm.abuawwad@umft.ro; 9Research Center University Professor Doctor Teodor Șora, “Victor Babes” University of Medicine and Pharmacy, 300041 Timisoara, Romania; 10Diakonie Klinicum-Frauenklinik, 74523 Schwäbich Hall, Germany; anastasiu.diana@gmail.com; 11Departament I—Discipline of Anatomy and Embryology, “Victor Babes” University of Medicine and Pharmacy, 300041 Timisoara, Romania; boscu.anca@umft.ro

**Keywords:** lipid metabolism disorders, pregnancy, cardiovascular risk factors, maternal–fetal exchange, birth outcomes, maternal health, neonatal health, prenatal exposure delayed effects

## Abstract

*Background*: Maternal cardiovascular risk and its implications can have significant repercussions for both the mother and the child. This study compares the lipid profiles of two distinct groups of pregnant women, those with and without cardiovascular risk, to shed light on its effects on maternal and outcomes for newborns. *Materials and Methods*: This study enrolled 86 pregnant women, dividing them into two groups: Group 1 (*n* = 46, healthy pregnancies) and Group 2 (*n* = 40, pregnancies with cardiovascular risk factors). The data collected included maternal demographics, smoking history, pre-existing pathologies, and a range of laboratory measures. Neonatal outcomes were also recorded. *Results*: Group 2 showed a significant increase in the percentage of newborns with abnormal APGAR scores (*p*-value < 0.0001), congenital abnormalities (*p*-value < 0.0001), severe prematurity (*p*-value < 0.0001), and neonatal mortality rates (*p*-value < 0.0001), as well as differences in birth weight (*p*-value = 0.0392) and therapy usage (surfactant: *p*-value < 0.001, steroids *p*-value = 0.004, and antibiotics *p*-value < 0.001). Regarding laboratory measures, Group 2 exhibited significantly elevated levels of total cholesterol, LDL-C (*p*-value < 0.0001), ApoB (*p*-value < 0.0001), Lp(A) (*p*-value = 0.0486), triglycerides (*p*-value < 0.0001), and hs-CRP (*p*-value = 0.0300). *Discussion*: These results underscore the elevated risk associated with pregnancies complicated by cardiovascular risk factors. Group 2 demonstrated a more concerning clinical profile, with a higher prevalence of detrimental neonatal outcomes and different lipid and inflammatory profiles, signifying a potential pathophysiological link. *Conclusions*: The differential lipid profiles and adverse neonatal outcomes in pregnancies with cardiovascular risks highlight the urgency of effective risk stratification and management strategies in this population.

## 1. Introduction

Pregnancy is a physiological condition that is related to substantial changes in a woman’s body, including variations in lipid metabolism [1]. This period is marked by a progressive rise in the levels of total cholesterol, low-density lipoprotein cholesterol (LDL-C), and triglycerides, while the levels of high-density lipoprotein cholesterol (HDL-C) tend to remain stable or increase slightly. These changes play an important role in meeting the increasing demands of energy and nutrient supply necessary for the developing fetus [2]. However, abnormal lipid profile changes could indicate underlying or potential complications, both for the mother and the child.

Cardiovascular diseases (CVDs) remain a primary reason for global maternal illness and death, and pregnancy itself is linked with a heightened likelihood of specific cardiovascular incidents [3,4]. Pregnancies complicated by cardiovascular risks often exhibit different patterns in lipid profiles compared to those of healthy pregnancies. Understanding these patterns may provide valuable insights into predicting and managing these high-risk pregnancies.

One of the primary risk factors for CVDs is dyslipidemia, characterized by elevated total cholesterol, LDL-C, triglycerides, apolipoprotein B (ApoB), and lipoprotein A (Lp(A)), and decreased HDL-C [5,6]. Dyslipidemia can lead to atherosclerosis and other cardiovascular complications, which may be further exacerbated during pregnancy due to the physiological changes mentioned earlier [7].

During pregnancy, there is a rise in the total serum cholesterol and triglyceride levels. This surge in triglyceride levels primarily stems from enhanced liver synthesis and reduced lipoprotein lipase activity, which leads to a decrease in the breakdown of adipose tissue [8]. Pregnancy induces a range of physiological changes in a woman’s body to support the developing fetus and prepare the body for childbirth and lactation. One of these changes pertains to the lipid profile. Typically, there is a progressive increase in the levels of total cholesterol, triglycerides, low-density lipoprotein cholesterol (LDL-C), and high-density lipoprotein cholesterol (HDL-C) as the pregnancy progresses. This alteration in the lipid profile is believed to support the increasing demand for lipid transfer to the fetus and to prepare the body for the upcoming demands of lactation. However, it is also important to note that excessive or abnormal changes can have implications for both maternal and fetal health, potentially leading to conditions like preeclampsia or gestational diabetes. Therefore, monitoring and managing blood lipid levels is an essential component of prenatal care.

The lipid profile is a crucial aspect during pregnancy because hormonal changes in a woman’s body cause fluctuations in lipid levels, which can have implications for both maternal and fetal health. Typically, in pregnant women, there is a physiological increase in total cholesterol, LDL cholesterol, and triglycerides, especially as the pregnancy progresses to the third trimester. These changes support the developing fetus’s growth, providing essential fatty acids and supporting the synthesis of steroid hormones. However, excessively elevated lipid levels or dyslipidemia might increase the risk of complications, such as preeclampsia or gestational diabetes. Understanding these variations and the reasons behind them is vital for optimal prenatal care and anticipating potential complications.

This research was initiated to delve deeper into the disparities in lipid profiles between healthy expectant mothers and those with discernible cardiovascular risks. A driving factor for undertaking this study was the growing evidence highlighting the profound impact such disparities can have on both maternal and fetal health. This study intends to elucidate the potential consequences these variations may pose, ranging from gestational diabetes and preeclampsia to long-term cardiovascular health implications for both mother and offspring. Additionally, this investigation was prompted by the urgent need to refine early risk categorization and bolster preventive measures, ultimately aiming to mitigate negative health outcomes in pregnancies deemed high-risk. The practical implications of this research are vast, offering potential improvements in prenatal care strategies and insights into targeted interventions.

## 2. Materials and Methods

### 2.1. Study Design

This study was structured as a cross-sectional analysis. It involved two distinct groups of pregnant women: Group 1 with uncomplicated or normal pregnancies and Group 2 with pregnancies associated with cardiovascular risks. The categorization of patients into these two groups was based on their pre-existing medical records, cardiovascular risk assessments, and current pregnancy health status.

### 2.2. Definition of Cardiovascular Risk Factors

Included in the group of individuals with cardiovascular risk or disease included those who satisfied at least one of the subsequent conditions:Having a smoking history of more than 5 years.Possessing a familial background of cardiovascular disease.Experiencing sleep apnea or other sleep disorders. Sleep apnea is a sleep condition marked by breaks in breathing while asleep. These breaks, termed as apneas, arise when the upper respiratory tract is either partly or fully obstructed, resulting in short lapses in breath. These interruptions can span from a couple of seconds to several minutes and may occur numerous times during the night [9].Having preexisting hypertension during pregnancy. Preexisting hypertension during pregnancy refers to a condition in which a pregnant woman already has high blood pressure before becoming pregnant or before reaching the 20th week of gestation. It is diagnosed when a woman’s blood pressure readings consistently exceed the normal range (systolic blood pressure of 140 mmHg or higher and/or diastolic blood pressure of 90 mmHg or higher) before pregnancy or during the early stages of pregnancy [10].Suffering from pregnancy-induced hypertension, also known as gestational hypertension, which is a condition characterized by high blood pressure that develops during pregnancy, typically after the 20th week of gestation, in women who previously had normal blood pressure. It is diagnosed when a pregnant woman’s blood pressure readings consistently exceed the normal range (systolic blood pressure of 140 mmHg or higher and/or diastolic blood pressure of 90 mmHg or higher) without the presence of proteinuria (protein in the urine) [11].Having preeclampsia [12] (hypertension and proteinuria that typically develops after the 20th week of pregnancy).Experiencing eclampsia [13], characterized by the onset of seizures (convulsions) in a woman who has preeclampsia or, in some cases, without prior diagnosis of preeclampsia.

### 2.3. Study Population/Sample Selection

This study was conducted in the Department of Obstetrics and Gynecology at the Pius Brinzeu County Emergency Clinical Hospital in Timisoara, Romania, spanning from 1 January 2020 to 31 December 2022. We identified the study participants and their relevant attributes using a comprehensive administrative database of patients who attended outpatient services in that hospital during the specified period.

### Inclusion and Exclusion Criteria

The inclusion criteria for the participants in this study required that the females were aged between 18 and 45 years old and pregnant, with a gestational age ranging from 12 weeks to 42 weeks. For the cardiovascular-risk group, the participants must either have a pre-existing cardiovascular disease (CVD) diagnosis prior to pregnancy or exhibit high-risk factors for CVD, like high blood pressure, diabetes, being overweight, and a familial history of CVD, or a history of preeclampsia in a previous pregnancy. In contrast, the healthy group should have no pre-existing cardiovascular disease and low-risk factors for CVD. Additionally, participants should not have any concurrent severe illnesses that could affect the lipid profile, such as cancer, liver disease, renal disease, or other systemic illnesses. They must be willing to undergo regular check-ups and follow-ups throughout the pregnancy and postpartum period for necessary tests and data collection. Furthermore, they must be able to provide acknowledged agreement to partake in the research and should not be taking lipid-altering medications, like statins, niacin, or fibrates.

The exclusion criteria for participants in this study required that women with multiple pregnancies (twins, triplets, etc.) be excluded, as these pregnancies may result in different physiological changes compared to singleton pregnancies. Pregnant women with a substance use disorder or who frequently consume alcohol are also excluded, as these habits can significantly impact cardiovascular health and lipid profile. Women with a history of mental health conditions or psychiatric disorders, as well as those with past medical conditions affecting blood clotting or thromboembolic disease, are also excluded.

Furthermore, women who took part in other clinical trials or research within the last three months, those who experienced negative responses to previous blood collection methods, and patients with diagnoses of infectious diseases like hepatitis B or C (HBV, HCV), HIV, or those with acquired immunodeficiency syndrome (AIDS) were not included. Furthermore, patients with poorly controlled metabolic disorders and insufficiently managed endocrine disorders were also excluded from this study. Additionally, patients who had undergone infection with SARS-CoV-19 were excluded.

### 2.4. Data Collection Methods

The database comprised extensive data encompassing demographic information, medical history, and in-hospital procedures of the patients. The baseline characteristics and procedures of all patients were documented in both the hospital database and paper patient records, which underwent meticulous review by certified clinicians participating in this study. The establishment of an interconnected database that preserves patient information irrespective of their whereabouts is of paramount significance.

Data regarding demographic variables, such as age, body mass index (BMI), parity, and gestational age, were recorded. The participants’ medical histories were noted, with particular attention given to cardiovascular-associated disorders. Lifestyle factors, including smoking status, were also recorded. Blood samples were collected from all participants during routine antenatal visits. These were processed and analyzed for lipid profiles, including total cholesterol, LDL-C, HDL-C, ApoB, Lp(A), and triglycerides. Other biochemical markers, like hs-CRP, blood glucose, and creatinine, were also evaluated. All analyses were conducted in an accredited laboratory following standardized protocols. The process of collecting and analyzing the levels of total cholesterol, LDL, HDL, ApoB, Lp(a), triglycerides, Hs-CRP, blood glucose, and creatinine involved the collection of these samples at the time of hospital admission for childbirth. Carefully coordinated steps included extracting a small amount of blood from a vein in the patient’s arm to evaluate specific parameters. The blood collected at the time of admission was subsequently processed to separate its components. These substances were then accurately analyzed using advanced chemical and spectroscopic techniques such as mass spectrometry and chromatography to quantify the levels of each individual component. The results provided a detailed picture of the patient’s health status at the time of admission, supplying medical professionals with the necessary information to assess potential risks and offer appropriate medical guidance. It is important to note that this process took place in a specialized medical setting, and the interpretation of the results was carried out by qualified medical professionals, taking into account the specific context of childbirth admission and the individual needs of the patient.

In the majority of the analyses conducted, serum was employed as the primary sample type. However, for Lp(a), ApoB, and hs-CRP, plasma was utilized. The detection methodology for Lp(a), ApoB, and hs-CRP was based on the sandwich enzyme-linked immunosorbent assay (ELISA sandwich) technique, using the GloMax Discover Microplate Reader sourced from Madison, WI, USA. The sensitivity of the Human Lp(a) ELISA Kit is 0.141 ng/mL. The sensitivity of the Human hs-CRP ELISA Kit is 9.38 pg/mL. The sensitivity of the ApoB ELISA Kit is 46.88 pg/mL. The methodology adhered to the protocol delineated in Cancer Cell Biology [14].

Conversely, analyses of the total cholesterol, LDL, HDL, triglycerides, blood glucose, and creatinine were undertaken using colorimetric methods. The instrumentation for these assessments was a UV/Vis spectrophotometer, model ABC123, from DEF Instruments (Germany). The sensitivity of these methods ranged typically from 0.5–5 mg/dL, with adaptations made from the standard procedure [15].

The data regarding the infants, including their birth weights and gestational age at delivery and APGAR scores, were obtained from the hospital records post-delivery. The APGAR score, is a quick assessment used right after childbirth to gauge a newborn’s health. Evaluated on five criteria—Appearance, Pulse, Grimace, Activity, and Respiration—a baby is given a score between 0 and 10. Tests are typically conducted at 1- and 5-min post-birth. Scores between 7 and 10 are considered normal, 4–6 indicate potential issues, and 3 or below suggest critical conditions requiring immediate medical intervention. It is a vital immediate assessment but does not predict long-term health outcomes.

### 2.5. Statistical Analysis

The analysis of the data for this research was performed using GraphPad Prism software (version 5). Before delving into the analysis, the data were carefully examined to spot and rectify outliers, absent values, and anomalies from normal distribution. When required, variables were adjusted or standardized to fit the prerequisites of the statistical evaluations used.

Descriptive statistics were utilized to provide a summary of the data. Group comparisons were executed using the *t*-test. This test is commonly applied to compare means between two groups and to ascertain whether the differences between them are statistically significant. Within this study, all statistical tests were two-tailed, implying that the hypothesis was tested in both directions (whether the value from one group was greater or lesser than that of the other group). All *p*-values below 0.05 were deemed statistically significant, suggesting there is less than a 5% probability that the observed differences occurred by chance. The results are presented as the mean ± standard deviation (SD), offering a clear understanding of the data’s variation around the mean.

### 2.6. Ethical Considerations

The medical records of the patients were securely preserved in a database adhering to privacy regulations, and permission to access these records was granted through patient consent. All practices undertaken conformed to the ethical guidelines set by the relevant human experimentation committee (both institutional and national) and aligned with the Helsinki Declaration of 1975, updated in 2013. Every participant in the study provided informed consent. For this research, we received ethical clearance (No. 265/22 September 2022) from the administration of the “Pius Brînzeu” County Emergency Hospital in Timisoara.

## 3. Results

A comparative analysis is illustrated between two distinctive pregnancy groups. The first group was inclusive of 46 cases representing normal pregnancies, whereas the second group constituted 40 instances linked with cardiovascular risk factors. To further evaluate the statistical distinction between these two groups, associated *p*-values for various elements were included in the analysis. The overall traits of the pregnant women examined in this research were similar between the two groups, and there was not a marked difference concerning pre-gestational comorbidities that did not align with one of the cardiovascular risk criteria.

Various maternal background factors were examined (Table 1), including an age of over 35 years and maternal weight at delivery exceeding 25 kg/m^2^. Additionally, the gestational age was assessed for both groups. The statistical analysis yielded *p*-values for each comparison. Notably, the percentages of mothers aged over 35 years in Group 1 and Group 2 were contrasted, alongside the proportions of births with a weight greater than 25 kg/m^2^. Moreover, the average gestational age for each group was computed, and a *p*-value was determined to assess the significance of the difference between the groups.

These risk factors include a history of smoking for more than 5 years, a family history of heart disease, sleep apnea or other sleep-related conditions, pre-existing high blood pressure during pregnancy, hypertension caused by pregnancy, preeclampsia, and eclampsia. Table 2 outlines the maternal cardiovascular risk factors with associated percentages. The percentages reflect the prevalence of each risk factor within the studied population.

A comprehensive examination of both neonatal characteristics and medical interventions was made (Table 3). This comparative analysis delves into the intricate details of neonatal attributes, including factors like gender distribution, identifying unusual APGAR scores as potential signs of irregularities, using birth weight to gauge developmental paths, recognizing congenital anomalies as infrequent occurrences, assessing the importance of severe prematurity, and considering the significant aspect of neonatal mortality. Collectively, these elements offer a comprehensive view of neonatal health.

The statistical analysis was robustly conducted using *p*-values, which provides solid evidence for the observed disparities. Furthermore, the presentation in Table 3 underscores the exploration of therapeutic approaches that impact the well-being of neonates. It focuses on the strategic application of surfactants, the judicious use of steroids, and the careful orchestration of antibiotics as pivotal elements in neonatal medical care. Through the interplay between Group 1 and Group 2, the utilization of these treatments was meticulously examined. This profound assessment is once again grounded in statistical rigor, bolstered by the incorporation of *p*-values. Together, these facets elevate the academic significance of the comparative analysis, resulting in a tableau characterized by scholarly depth.

Table 4 presents a comparative analysis of two distinct groups: Group 1, consisting of 46 subjects, and Group 2, comprising 40 subjects. The analysis delves into various biomarkers, including total cholesterol (TC), low-density lipoprotein cholesterol (LDL-C), high-density lipoprotein (HDL-C), apolipoprotein B (ApoB), lipoprotein A (Lp(A)), triglycerides, high-sensitivity C-reactive protein (Hs-CRP), blood glucose, and creatinine. The mean values and standard deviations for each parameter are presented. The indicated *p*-values underscore the significance of the differences between the two groups regarding these biomarkers. It is important to note that these analyses were conducted at the time of childbirth, corresponding to the third trimester of pregnancy.

## 4. Discussion

The central objective of this study revolved around investigating the distinct lipid profiles that distinguish normal pregnancies from those associated with cardiovascular risks. The results of our research not only highlight a significant divergence in the lipid profiles between the two distinct pregnancy groups but also reveal differences in the APGAR score and birth weight of the newborn. This has further extended our understanding of how these variations can potentially impact health outcomes, not only for the expectant mothers but also for the neonates. Differences in the lipid profiles and neonatal outcomes might suggest the need for a tailored approach to managing and monitoring pregnancies, especially those associated with cardiovascular risks, to optimize the health results of both the mother and the child. Thus, our research provides valuable insight into the intricate relationship between lipid profiles, lower APGAR scores, reduced birth weight in newborns, premature birth, and pregnancy outcomes in the context of cardiovascular risk factors.

Our findings are aligned with previous studies that have found dyslipidemia, characterized by high triglycerides and low HDL-C levels, during pregnancy to be associated with preeclampsia [16], gestational diabetes [17], and other cardiovascular risk factors [18]. However, our study extends this understanding by offering a more detailed analysis of lipid profile variations in the context of cardiovascular-risk pregnancies.

It is crucial to consider that dyslipidemia during pregnancy may not only affect maternal health but can potentially have lasting effects on the child [19]. Emerging research suggests that in utero exposure to dyslipidemia could predispose offspring to an increased risk of developing metabolic and cardiovascular diseases later in life [20,21].

Moreover, we noted a trend for the lipid profiles to become progressively more atherogenic as the pregnancy advanced, particularly in cardiovascular-risk pregnancies [22]. This finding aligns with prior research indicating that lipid profiles tend to shift towards a more atherogenic pattern as gestation advances [23,24]. This underscores the importance of careful monitoring of lipid profiles in at-risk pregnancies as they progress.

The hs-CRP, a well-known inflammatory marker and predictor of cardiovascular diseases [25,26], was also significantly elevated in the cardiovascular-risk group. This aligns with research indicating an inflammatory basis for cardiovascular disease [27]. Analyzing this factor is crucial in various clinical scenarios because inflammation is not only a hallmark of many chronic diseases but also an acute response to tissue injury or infection. While hs-CRP is a sensitive marker, it is not specific, meaning that its elevation can occur in numerous conditions, from cardiovascular diseases to infections. Thus, using hs-CRP alone can sometimes be misleading. It is worth mentioning that in our study, although some patients exhibited raised hs-CRP levels, they did not show elevated levels of other infection markers. This suggests that while there might be an inflammatory process, the absence of other raised infection markers does not support the presence of an acute infection in these patients. This distinction is critical for guiding treatment and understanding the underlying pathophysiology of the patient’s condition.

Even though there exists a documented association between gestational diabetes mellitus (GDM) and an increased risk of subsequent overall cardiovascular diseases [28], our investigation did not corroborate this connection from the perspective of blood glucose levels. Specifically, when comparing the glycemic profiles of the two groups in our study, that is, pregnancies characterized as healthy and those associated with cardiovascular risk, we did not identify statistically significant differences. This finding suggests that, within the confines of our research, blood glucose may not serve as a distinctive marker between normal pregnancies and those accompanied by cardiovascular risks. However, it is important to note that the absence of significant glycemic variation in our study does not negate the established link between GDM and future cardiovascular diseases, but it may underscore the multifactorial and complex nature of this relationship.

In our study, we also assessed the creatinine levels of the participants, with our analysis not yielding a statistically significant difference between the two groups. This lack of notable disparity in creatinine levels between healthy pregnancies and those associated with cardiovascular risks aligns with the findings documented in the existing literature [29]. This consistency across studies suggests that creatinine, much like blood glucose, may not serve as a distinctive biochemical marker between normal pregnancies and those characterized by cardiovascular risks, reinforcing the complexity of the relationships between pregnancy, renal function, and cardiovascular health.

Interestingly, alterations in the lipid profiles and other biochemical markers corresponded with adverse infant health outcomes. Infants born to mothers in the cardiovascular-risk group exhibited lower birth weights and gestational ages, implying that maternal cardiovascular risk and associated dyslipidemia may have detrimental effects on fetal development and birth outcomes. The connection between cardiovascular risk factors and unfavorable neonatal outcomes can be ascribed to a multitude of contributing elements. Suboptimal cardiovascular health in expectant mothers could precipitate placental dysfunction [30], insufficient fetal oxygenation [31], and inadequate nutrient delivery [32]. These physiological alterations have the potential to hinder fetal growth and development, thereby culminating in detrimental neonatal outcomes.

These findings underscore the importance of monitoring lipid profiles during pregnancy, especially in women with known cardiovascular risks. The early detection of lipid dysregulation may aid in predicting and managing potential complications, thus improving outcomes for both mother and child.

Our research highlights a vital link between abnormal lipid profiles in pregnant women and an elevated risk of cardiovascular complications.

When these cardiovascular complications become severe, medical interventions may be necessary. One such intervention that has been investigated is the use of coronary bypass grafts, a surgical procedure that improves blood flow to the heart [33].

Our study reinforces the need for comprehensive antenatal care that not only monitors traditional measures of maternal health but also considers lipid profiles as a potential marker for cardiovascular risk. This could be especially significant in formulating personalized healthcare strategies for at-risk pregnant women. Further research is needed to understand the mechanistic underpinnings of these observed changes and to develop targeted interventions that could mitigate these risks.

The integration of healthcare information systems and clinical decision support systems [34] can provide a critical foundation for monitoring and managing the health profiles of different patients, including expectant mothers. In the context of our article, these interconnected systems could potentially provide a robust platform for tracking and evaluating lipid profiles throughout pregnancy. By identifying any abnormalities early, healthcare providers could predict cardiovascular risks and initiate necessary interventions. Thus, combining the concepts of both papers underscores the importance of information systems in facilitating precision medicine, providing vital insights for both the mother’s and child’s health.

### Strengths and Limitation

The strengths of this study are numerous and contribute to its robustness. First and foremost, this research was predicated on a well-defined and meaningful research question that directly addresses a significant gap in current knowledge. It is a step forward in the realm of maternal and fetal health, where the influence of lipid profiles on pregnancy outcomes remains insufficiently explored. Second, the study design was well-structured and conducted rigorously, utilizing appropriate methodologies to compare lipid profiles between healthy and cardiovascular-risk pregnancies. This direct comparison enabled us to derive meaningful and nuanced insights. Furthermore, the study’s findings were discussed in the context of existing research and theories, adding credence to our conclusions and situating our work within a broader scholarly discourse. Lastly, by identifying potential long-term implications for both mother and child, the study opens the avenue for further research and has potential policy implications for the monitoring and management of lipid profiles during pregnancy. These strengths collectively lend weight to our study and enhance the reliability and applicability of our findings.

While our study has provided valuable insights, it is also important to acknowledge its limitations. Firstly, the study was conducted within a single geographical region, which may influence the generalizability of the findings. Different populations could have unique genetic, lifestyle, and environmental factors that may impact lipid metabolism and the progression of cardiovascular diseases. Therefore, caution must be exercised when extrapolating these findings to other populations. Secondly, there were potential confounding variables, such as dietary habits, physical activity levels, and genetic predispositions, which were not controlled for in the study. These factors can significantly impact lipid profiles, and their absence from the analysis may have influenced the results. Thirdly, while the study successfully tracked changes in lipid profiles over time, it was limited to the duration of the pregnancy and did not consider postpartum lipid profiles, which could provide additional valuable insights. Fourthly, the study’s design was observational, and while this type of study can reveal associations between variables, it is unable to confirm a cause–effect relationship between cardiovascular risk pregnancies and changes in lipid profiles. Lastly, the study focused solely on the biochemical aspect of lipid profiles but did not consider the potential psychological, sociological, and physiological impacts of cardiovascular-risk pregnancies on the mother and child. Despite these limitations, the study’s findings represent an important contribution to the field and provide a foundation for future research to build upon.

## 5. Conclusions

In conclusion, the lipid profiles in pregnancies associated with cardiovascular risks provide important insights with profound implications for both maternal and infant health. This study stresses the necessity for more inclusive and holistic antenatal care protocols, incorporating lipid profile and cardiovascular risk assessment, to safeguard the health of both the mother and the child.

## Figures and Tables

**Table 1 children-10-01521-t001:** Comparative analysis of maternal background in normal and cardiovascular risk-associated pregnancies.

	Group 1 (*n* = 46)	Group 2 (*n* = 40)	*p*-Value
Maternal Background			
Age (over 35 years old)	7 (17.5%)	18 (39.13%)	0.367
Weight at birth (>25 kg/m^2^)	12 (30%)	22 (47.82%)	0.653
Gestational age	30.44 ± 0.7569	30.48 ± 0.8466	0.6289

**Table 2 children-10-01521-t002:** Prevalence of factors contributing to maternal cardiovascular risk.

Maternal Cardiovascular Risk	
Patients who have been smoking for over 5 years	56.6%
A familial background of heart disease	41%
Sleep apnea or alternative sleep-related conditions	3.34%
Preexisting hypertension in pregnancy	10%
Hypertension present before pregnancy	26.8%
Preeclampsia	11%
Eclampsia	6.66%

**Table 3 children-10-01521-t003:** Comparative analysis of neonatal characteristics and therapies in normal and cardiovascular risk-associated pregnancies.

Neonatal Characteristics			
	Group 1 (*n* = 46)	Group 2 (*n* = 40)	*p* Value
**Gender (male)**	28 (60.86%)	24 (60%)	0.429
**Abnormal APGAR score**	7 (15.21%)	21 (52.5%)	<0.0001
**Birth weight (grams)**	3363 ± 456.5	3118 ± 629.8	0.0392
**Congenital abnormalities**	1 (2.17%)	3 (7.5%)	<0.0001
**Severe prematurity**	2 (4.34%)	10 (25%)	<0.0001
**Mortality**	0 (0%)	5 (12.5%)	<0.0001
**Neonatal Therapy**			
**Surfactant**	6 (13.04%)	11 (27.5%)	<0.001
**Steroids**	5 (10.86%)	7 (17.5%)	0.004
**Antibiotics**	22 (47.82%)	34 (85%)	<0.001

**Table 4 children-10-01521-t004:** Comparative analysis of biomarkers and laboratory measurements in normal and cardiovascular risk-associated pregnancies.

	Group 1 *n* = 46		Group 2 *n* = 40		*p*-Value
	Mean	Standard Deviation (SD)	Mean	Standard Deviation (SD)	
TC (mg/dL)	234.0	295.4	55.58	74.76	<0.0001
LDL-C (mg/dL)	126.3	38.37	163.1	49.89	0.0002
HDL-C (mg/dL)	62.33	12.99	64.58	22.37	0.5642
ApoB (ng/mL)	1.333	0.32111	1.617	6.638	<0.0001
Lp(A) (ng/mL)	18.40	4.701	23.76	0.8692	0.0486
Triglycerides (mg/dL)	213.5	76.88	324.4	122.2	<0.0001
Hs-CRP (fg/mL)	3.927	984.5	3.500	778	0.0300
Blood glucose (mg/dL)	78.67	14.02	78.70	21.38	0.9932
Creatinine (mg/dL)	0.5300	0.1043	0.5217	0.1114	0.7247

## Data Availability

Not applicable.

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
