# Peer review of "Lipid Profile Variations in Pregnancies with and without Cardiovascular Risk: Consequences for Both Mother and Newborn"

_children, 2023, doi:10.3390/children10091521_

Round 1

Reviewer 1 Report

The authors represent the results of the moncentric, retrospective study regarding the comparison of outcomes of women without cardiovascular risk factors and pregnancy with a cohort with cardiovascular risk factors and as well relevant changes of the lipid profile. The alterations of the lipid profile are in line with present cardiovascular risk factors. The important point which the authors could show in their study is the detrimental effect of present risk factors, cardiovascular disease and thus altered lipid profile with the outcome of the new born. The study design is monocentric, however the patients are thus well characterized and show as well all a lab profile. The shown tables underline the message of the manuscript to the reader. In all, there are no additional points the authors have to attach to the manuscript.

Author Response

Thank you for taking the time to review our manuscript, detailing the outcomes of women with and without cardiovascular risk factors during pregnancy and the subsequent changes in their lipid profiles.

We are glad to note that you appreciate the depth and detailed characterization that our monocentric approach has facilitated. Indeed, we believe that while the focus was on one center, the comprehensiveness of the patient data and lab profiles has allowed for meaningful and insightful results.

We concur with your observation about the significance of demonstrating the detrimental effect of present risk factors, cardiovascular diseases, and the altered lipid profile on neonatal outcomes. The tables were indeed incorporated with an intent to make the findings more lucid and to strengthen the narrative for our readers.

Your feedback, particularly the acknowledgment that no additional points are required to enhance the manuscript, is encouraging. This gives us confidence in the clarity and completeness of our research presentation.

Once again, we appreciate your constructive insights and look forward to seeing our work published and contributing to the larger scientific discourse.

Warm regards,

Simona-Alina Abu-Awwad.

Reviewer 2 Report

Please, follow my suggestion step by step as indicated in the file in the attachment

Author Response

Firstly, I want to express my gratitude for the time you've dedicated to reviewing the manuscript. Your feedback has been instrumental in refining the article, and I have made the necessary changes in light of your suggestions:

  1. I've incorporated a detailed description concerning the alterations of the lipid profile during pregnancy, aiming to provide readers with a comprehensive understanding.

  1. With regard to SARS-CoV-2 infected patients, I clarified any potential ambiguities by explicitly stating that patients who had undergone this infection were initially excluded from the study. This has now been highlighted in the text.

  1. As for patients with gestational diabetes, they were not excluded from the two groups. However, it's worth noting that none of the patients diagnosed with gestational diabetes met the inclusion and exclusion criteria for the study. Regarding the neonates, only two fetuses were sonographically estimated to be macrosomic, but at birth, they were within the normal weight parameters. The other newborns were neither sonographically estimated as macrosomic nor did they have birth weights that categorized them as such.

  1. Regarding the symbol you pointed out, it was indeed unintentional. Originally, something else was in its place, and its presence now was an oversight. I've removed it to maintain clarity in the manuscript.

  1. Concerning the hs-CRP, I've added a thorough description and analysis of this factor in the text, ensuring its significance and implications are clear.

  1. As for the study you've referenced, I couldn't find analogous data in the mentioned paragraph for a direct comparison. However, if you believe it's beneficial, I'm more than willing to draw a comparison with that study in another section of the article.

Once again, I appreciate your insightful feedback and hope that these revisions make the manuscript more robust and coherent.

Reviewer 3 Report

The manuscript entitled “ Lipid Profile Variations in Pregnancies with and without Cardiovascular Risk: Consequences for both Mother and Newborn” The manuscript needs substantial revision.

-        “The interplay between pregnancy and maternal cardiovascular risk is complex” , This should be modified because it gives impression that pregnancy is the cause of cardiovascular risk in general.

-        Line 38 :” APGAR” ?

-        “ a significantly higher” and “ significantly higher”, should be substantially adjusted for language edits and revised, plz also add (the P value).

-        Line 80: “Pregnancy involves numerous physiological changes, including alterations in the lipid profile [8]” was mentioned with the same way in the first paragraph of the introduction.

-        “monitoring and managing blood lipid levels is an essential component of prenatal care”: is it a routine for all pregnant women to monitor this, I don’t think so ?

-        In the introduction, please identify how lipid profile is a critical point, is it varied in normal pregnant women and why ?

-        Line 117: What is the differences between high blood pressure and hypertension, (high blood pressure (hypertension and proteinuria that 117 typically develops after the 20th week of pregnancy) revise this.

-        I think this range is very wide (18 and 45 years) also the percentage of pregnancy is less within 45 years and few women will get pregnant in this age especially when they were at risk of CVD.

-        What does this mean “ maternal weight at delivery exceeding 25 kg/m2.”

-        The method of detection of TC, HDL  and the data presented in table 4 were missed in the methods , how did you detect and how did you monitor  while you detected them as single time during pregnancy?

-        In the table footnote , no full names of parameters were presented and no P value or (N)?

-        Discussion needs revision and some paragraphs are of no value, you should compare the results with the previous researches.

-        The conclusion is too long , the last section only could be enough.

Moderate English language is required.

Author Response

The manuscript entitled “Lipid Profile Variations in Pregnancies with and without Cardiovascular Risk: Consequences for both Mother and Newborn” The manuscript needs substantial revision.

- “The interplay between pregnancy and maternal cardiovascular risk is complex”, This should be modified because it gives impression that pregnancy is the cause of cardiovascular risk in general.

Thank you for pointing out the potential implications of the phrasing. I've revised the statement to ensure it doesn't give the impression that pregnancy is the general cause of cardiovascular risk. I appreciate your feedback and aim for clarity in our communication.

-        Line 38:” APGAR”?

                  Thank you for your inquiry about the APGAR score. I have provided a detailed explanation of the APGAR score under the "Data Collection Methods" section. If you refer to that section, you will find all the necessary information about its significance and how it is assessed.

- “a significantly higher” and “significantly higher”, should be substantially adjusted for language edits and revised, plz also add (the P value).

I have made the necessary revisions to the text, replacing "a significantly higher" and "significantly higher". Additionally, the P value has been added as per your request. Please review the updated content to ensure it meets your expectations.

-        Line 80: “Pregnancy involves numerous physiological changes, including alterations in the lipid profile [8]” was mentioned with the same way in the first paragraph of the introduction.

Thank you for bringing this to my attention. I've revised the mentioned phrase at Line 80 to ensure that it doesn't repeat in the same manner as in the first paragraph of the introduction. Your vigilance is much appreciated, and I believe this change enhances the clarity and flow of the work.

- “monitoring and managing blood lipid levels is an essential component of prenatal care”: is it a routine for all pregnant women to monitor this, I don’t think so?

You are correct; monitoring blood lipid levels is not currently a routine practice for all pregnant women. However, in our opinion, these analyses should be performed at least once, especially for those with a history of dyslipidemia. We believe that such a step can offer valuable insights into maternal health and potential risks during pregnancy.

-        In the introduction, please identify how lipid profile is a critical point, is it varied in normal pregnant women and why?

I have added a paragraph in the introduction that specifically addresses the significance of the lipid profile during pregnancy. This addition elaborates on how lipid levels vary in typically pregnant women and the reasons behind these changes. I trust this enhancement will provide clarity and depth to the topic as you suggested.

-        Line 117: What are the differences between high blood pressure and hypertension, (high blood pressure (hypertension and proteinuria that 117 typically develops after the 20th week of pregnancy) revise this.

Thank you for pointing out the ambiguity on Line 117. I have revisited and clarified the content

-        I think this range is very wide (18 and 45 years) also the percentage of pregnancy is less within 45 years and few women will get pregnant in this age especially when they were at risk of CVD.

                  You are right; it is indeed a very wide range. We chose this age range to encompass a broad spectrum of pregnant women, from the younger ones to those approaching menopause. We aimed to have a holistic view on how the risk of CVD might vary with age and to ensure that we don't overlook any subcategory of women, no matter how small their percentage might be. We understand your concern and thank you for your feedback.

-        What does this mean “maternal weight at delivery exceeding 25 kg/m2.”

The statement "maternal weight at delivery exceeding 25 kg/m^2" refers to the body mass index (BMI) of the mother at the time of delivery. A BMI exceeding 25 kg/m^2 indicates that the mother is categorized as overweight at the time of delivery.

-        The method of detection of TC, HDL and the data presented in table 4 were missed in the methods , how did you detect and how did you monitor  while you detected them as single time during pregnancy?

I apologize for any confusion in the methods section. In the provided explanation, I included the process of collecting and analyzing the data at a single time during pregnancy. However, I must clarify that we did not monitor these parameters continuously; instead, we collected these samples once during hospital admission for childbirth. The method for detecting TC, HDL, and the data in table 4 were inadvertently missed in the methods section, and I appreciate you bringing this to my attention. We intend to conduct a comprehensive study in the future, encompassing monitoring from the first trimester through post-delivery, to gain a more comprehensive understanding of the changes in these parameters over the course of pregnancy. This longitudinal approach will provide valuable insights into the dynamics of these analytes throughout pregnancy.

-        In the table footnote, no full names of parameters were presented and no P value or (N)?

From my perspective adding the full names of parameters, the P value, and N for each parameter in the footnote would make the description overwhelmingly lengthy. However, if you believe that this addition would significantly enhance the quality or clarity of the article, I'm willing to make the adjustment.

-        Discussion needs revision and some paragraphs are of no value, you should compare the results with the previous researches.

                  Thank you for your feedback. I've revised the discussion section as you suggested and removed the paragraphs that didn't add value. I've also made sure to compare our results with previous researches to provide a more comprehensive perspective. Please let me know if there are any further adjustments you'd like me to make.

-        The conclusion is too long; the last section only could be enough.

I've taken your feedback into account and trimmed down the conclusion. I've retained only the last section, as you suggested, believing it captures the essence of our findings succinctly. Please review it and let me know if there are any further changes you'd like to see.

Round 2

Reviewer 3 Report

I thank the authors for the revision of the manuscript in this short time. However, there is still a point regarding the detection of  "The pro- 222 cess of collecting and analyzing levels of total cholesterol, LDL, HDL, ApoB, Lp(a), tri- 223 glycerides, Hs-CRP, blood glucose, and creatinine" from line 222 : No adequate methods were provided y. The author should present the type of samples " serum as you mentioned " , then each one of these parameters , how it was detected  " spectrophotometric or colorometric ? , the name of the apparatus used and origin? the type of kit used  ? their sensitivity and origin?  with the refrences for the methods followed. 

Minor typos mistakes.

Author Response

Thank you for bringing this to our attention. We have now addressed the point regarding the detection of "The process of collecting and analyzing levels of total cholesterol, LDL, HDL, ApoB, Lp(a), triglycerides, Hs-CRP, blood glucose, and creatinine" from line 222. We have provided the necessary details for each parameter, including the detection method, whether it was spectrophotometric or colorimetric, the name and origin of the apparatus used, the type of kit employed, its sensitivity, and its origin. Furthermore, we have added the appropriate references for each method followed. We appreciate your feedback and have made these updates in the revised manuscript.